# Modelling Method and Application of Anti-Corrosion Pill Particles in Oil and Gas Field Wellbore Casing Annulus Based on the Discrete Element Method

**Dongtao Liu [1,*], Chunshang Qiao [1], Jun Wan [2], Yuliang Lu [1], Jiming Song [1], Zhenhe Yao [1], Xinjie Wei [1] and Yajun Yu [3,*]**

1   CNOOC EnerTech-Drilling & Production Co., Shenzhen 518067, China; qiaochsh@cnooc.com.cn (C.Q.); luyl@cnooc.com.cn (Y.L.); songjm2@cnooc.com.cn (J.S.); yaozhh@cnooc.com.cn (Z.Y.); weixj6@cnooc.com.cn (X.W.)
2   Shenzhen Branch, CNOOC (China) Co., Ltd., Shenzhen 518067, China; wanjun2@cnooc.com.cn
3   School of Biological and Agricultural Engineering, Jilin University, Changchun 130022, China
\*   Correspondence: liudt@cnooc.com.cn (D.L.); yuyajun@jlu.edu.cn (Y.Y.); Tel.: +86-138-2879-7030 (D.L.); +86-151-4317-3701 (Y.Y.)

**Abstract:** This study uses a self-developed anti-corrosion pill particle as the research object and develops the pill particle population modelling method in order to optimize the anti-corrosion process of oil and gas wellbore casing annuli. The shape of the pill particle is similar to a cylinder, according to the test and analysis of geometrical characteristics, and can be simplified into three types based on height, namely pill particles A (5.4 mm), B (5.8 mm), and C (6.2 mm). The multi-sphere approach is then used to create models of three different types of pill particles with varying degrees of precision. The feasibility and effectiveness of the modelling method for pill particle populations are proven by comparing the simulation results of the bulk density test and the angle of repose test. The results show that the 12-sphere models of pill particles A, B, and C are accurate representations of genuine pill particle morphologies and are adequate for simulating particle mechanics and flow processes. The applicability and practical use of the modelling method are then demonstrated using an example of a self-designed pill particle discharging mechanism. The results show that the modelling method can accurately simulate the pill discharging process and provide an accurate simulation model and theoretical basis for the optimization of the structural parameters, dimension parameters, and operating parameters of the discharging device.

**Keywords:** discrete element method; particle modelling; multi-sphere method; simulation analysis; anti-corrosion pill particle

## 1. Introduction

The anti-corrosion of oil and gas field wellbore casing annuli is a critical link in the integrity management of oil and gas field wellbores. Traditional anti-corrosion fluid perfusion is harmed by the tiny area and buildup of oil sludge, and it has a number of disadvantages, including difficulties in lowering the pipeline and restricted anti-corrosion fluid perfusion. Our organization has created a solid slow-release anti-corrosion pill particle for this purpose. However, due to the complexity of the working conditions, such as a single route for pill discharging and a small inner diameter space for the casing, optimizing the pill discharging device to minimize clogging and fragmentation of the pill discharging process and to increase its stability and uniformity has become a critical technical bottleneck to overcome [1–3]. The discrete element method (DEM) [4], based on particle dynamics, is capable of analyzing the interaction between particles and between particles and mechanical components from a microscopic perspective, obtaining microscopic data such as particle displacement, velocity, acceleration, and so on, and revealing the influencing

factors and working mechanism of mechanical components. Thus, DEM has developed into a potent tool for optimizing mechanism design and overcoming technological constraints in industrial and agricultural output [5–10].

Particle modelling, being the central issue in DEM, is a critical aspect in determining the accuracy of simulation results. Currently, the polyhedron method [11], super-quadric equation method [12], and multi-sphere method [13–15] are the most frequently utilized particle modelling methods. The multi-sphere approach is more extensively utilized in industrial and agricultural production due to the simplicity of the contact detection algorithm. For example, Danesh et al. [16] used a multi-sphere method to create ballast particle shapes to investigate the macroscopic and microscopic mechanics shear behavior of railroad ballast and demonstrated that the particle shape has an effect on the formation of shear bands via the microscopic mechanics response. Tekeste et al. [17] modeled the soil particles using a multi-sphere approach and developed a proportionate connection between the soil response force and the bulldozer blade length. Horabik et al. [18] constructed one-, three-, four-, and six-sphere models for wheat kernels using a multi-sphere method and demonstrated that the four-sphere model could better reproduce the transmission of particle–particle forces from the vertical to the lateral directions through loading and unloading cyclic compression tests. Zeng et al. [19] used the multi-sphere approach to construct a 16-sphere model of rice kernels and disclosed the rice crushing mechanism by the simulated study of the rice milling process. Due to the manufacturing process, the size of the pill particle generated by our organization varies. Thus, how to examine the shape and size characteristics of pill particles through testing, taking into account the unpredictability of their scale distribution, and establish a technique for modelling the pill particle population must be thoroughly investigated.

On the other hand, validation of the modelling method is a critical problem that must be addressed. At the moment, the majority of modelling approaches are verified by simulating static and dynamic experimental processes [20–22]. For example, Zhang et al. [23] utilized the multi-sphere technique to create four different kinds of soil particle models and then analyzed the operation of a deep loosening machine working on the soil using a combined DEM–MBD simulation, demonstrating the modelling method's utility. Tao et al. [24] employed a multi-sphere technique to construct three representations of ellipsoidal particles, and the modelling method's practicality was shown by moving bed experiments. Zhou et al. [25] used a multi-sphere method to model four different shapes of maize seed particles, including the horse-tooth shape, the truncated triangular pyramid shape, the ellipsoid cone shape, and the spheroid shape, and validated the proposed modelling method for maize seeds using the bulk density, the angle of repose, and self-flow screening. Sun et al. [26] used the multi-sphere method to construct a double-ellipsoidal 13-, 17-, 25-, and 33-sphere model for wheat seeds, as well as a single-ellipsoidal 5-, 9-, 13-, 17-, and 21-sphere model for wheat seeds. They validated the wheat seed particle model proposed in this paper by comparing experimental data to simulation results using the static angle of repose, self-flow screening, and dynamic angle of repose tests The proposed pill particle population modelling method requires an in-depth study of how to select an appropriate validation method to reveal the effect of the number of filled spheres on the particle population accumulation process and flow behavior, as well as to verify the modelling method's feasibility, validity, and applicability.

In order to address the aforementioned issues, this work uses self-developed anti-corrosion pill particles as the research object, measuring and analyzing their geometric form and dimension characteristics. On this premise, the multi-sphere approach was used to build particle models of three different forms of pill particles. The bulk density and angle of repose simulation results were compared to actual data to determine the feasibility and usefulness of the suggested approach for modelling the pill particle population in this work. Finally, the modelling method's applicability and practical use are shown using the self-designed pill discharge device as an example. The work presented in this article establishes an accurate simulation model and theoretical foundation for the future investigation of

the pill discharging process and optimization of the pill discharging device's structural characteristics, dimensional parameters, and operating parameters.

## 2. Materials and Modelling

### 2.1. Shape and Size Analysis

The self-developed annular air anti-corrosion pill particle is used as the research object, with 200 pill particles randomly chosen. The geometric forms and size characteristics of pill particles were determined, as well as the mass ratio of various shapes of pill particles. The findings indicated that the pill particle was approximately cylindrical in shape, as shown in Figure 1a, with a constant diameter (D) and a nearly uniform height (H) dispersed according to three scales. In the particle population, the mass ratios of pill particles A, B, and C are 35%, 35%, and 30%, respectively. Three scales of pill particles are modeled in this article using DEM and distributed according to their mass ratio in order to build a technique for modelling pill particle populations.

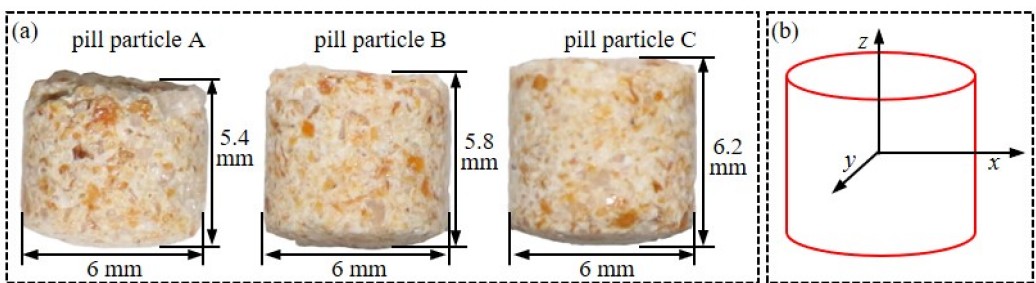

**Figure 1.** The form and coordinate system of pill particles: (**a**) Three different forms of pill particles and their associated size characteristics; (**b**) The pill particle's coordinate system.

### 2.2. Modelling Methods of Pill Particles

The pill particles are modeled in this article using a multi-sphere technique. Single-layer, double-layer, and triple-layer particles are used to characterize the pill particle form in order to analyze the influence of the number of filled spheres on the modeling accuracy and flow properties of the pill particles. The coordinate system of the pill particles is shown in Figure 1b, where the coordinate origin is the mass center of the particle, the x- and z-axis are the diameter and height directions of the particle, respectively, and the $y$-axis is defined by the right-hand rule.

#### 2.2.1. Modelling Methods for Pill Particle A and B

Because the particle diameter is smaller than the height for particles A and B, a 4-sphere model is generated using the particle diameter as the filling sphere's diameter. Based on the tangency and overlap of the particles, a 12-sphere double layer model and a 20-sphere triple layer model are created. Each layer of the triple layer model fills 6 spheres, with 2 more filled spheres in the centre of the top and bottom layers. The detailed modelling method is described below.

The point $(D/2-H/2, 0)$ in the xoz coordinate plane is filled by a sphere $O_1$ of radius $H/2$ and tangent to the cylindrical surface to the x axis, while spheres $O_2$ to $O_4$ are generated by an array of spheres $O_1$ arranged at a 90° angle around the z axis in the oxyz coordinate system, as shown in Figure 2a.

To minimize inaccuracy, the pill particle model was constructed in the z-axis direction using a double-layer filled sphere. At the point $(D/2H/3, H/6)$ in the xoz coordinate plane, a sphere $O_1$ with radius $H/3$ and tangent to the cylindrical surface to the x axis fills the space; spheres $O_2$ to $O_6$ are formed by an array of spheres $O_1$ arranged at a 60° angle around the z axis in the oxyz coordinate system. The spheres $O_7$ to $O_{12}$ are filled with the mirror image of the spheres $O_1$ to $O_6$, forming the 12-sphere model of the pill particle A depicted in Figure 2b.

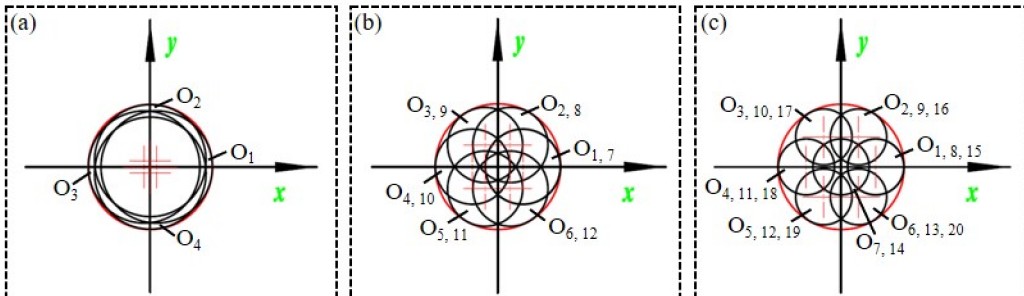

**Figure 2.** Diagram illustrating the procedure of filling a pill particle A: (**a**) A 4-sphere model; (**b**) A 12-sphere model; and (**c**) A 20-sphere model.

To further decrease inaccuracy, three layers of filled spheres were used to simulate the pill particles in the z-axis direction. At the point (D/2−H/4, H/4) in the xoz coordinate plane, a sphere $O_1$ with radius H/4 and tangent to the cylindrical surface to the x axis fills the space; spheres $O_2$ to $O_6$ are formed by an array of spheres $O_1$ arranged at a 60° angle around the z axis in the oxyz coordinate system. A sphere $O_7$ of radius H/4 and tangent to the cylindrical surface to the z axis is filled at the position (0, H/4) in the xoz coordinate plane. The spheres $O_8$ to $O_{14}$ are filled with the mirror image of the spheres $O_1$ to $O_7$ in the xoy coordinate plane. The point (D/2−H/4, 0) in the xoz coordinate plane is filled by a sphere $O_{15}$ with radius H/4 and tangent to the cylindrical surface to the x axis, while spheres $O_{16}$ to $O_{20}$ are generated by an array of spheres $O_{15}$ arranged at a 60° angle around the z axis in the oxyz coordinate system, as shown in Figure 2c.

As seen in Figure 3, the same filling procedure was employed to model the pill particles B.

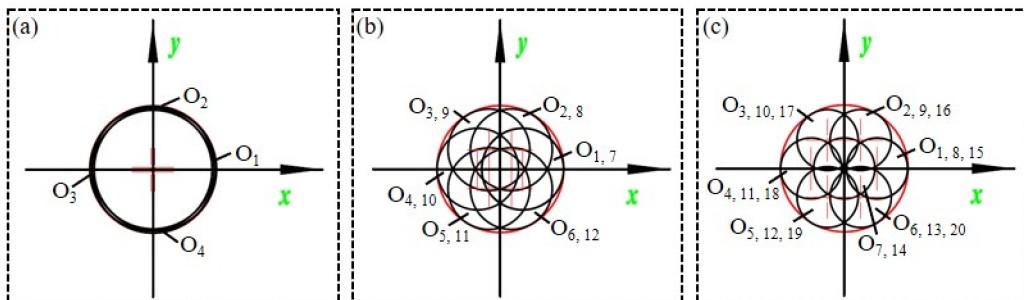

**Figure 3.** Diagram illustrating the procedure of filling a pill particle B: (**a**) A 4-sphere model; (**b**) A 12-sphere model; and (**c**) A 20-sphere model.

### 2.2.2. Modelling Methods for Pill Particle C

Because the particle diameter is greater than the height for particle C, a 2-sphere model with the particle height as the filled ball's diameter is developed. The tangency and overlap of the particles are the foundations of the 12-sphere double layer and 18-sphere triple layer models.

At the point (0, H/2−D/2) in the xoz coordinate plane, a sphere $O_1$ with radius D/2 and tangent to the cylindrical surface parallel to the *z* axis is filled; spheres $O_2$ are filled with the mirror image of spheres $O_1$ in the xoy coordinate plane, thereby constructing the two-sphere model of the pill particle C shown in Figure 4a.

At the point (D/2−H/3, H/6) in the xoz coordinate plane, a sphere $O_1$ with radius H/3 and tangent to the cylindrical surface to the *x* axis is filled; spheres $O_2$ to $O_6$ are formed by an array of spheres $O_1$ arranged around the *z* axis at an angle of 60° in the oxyz coordinate system. As illustrated in Figure 4b, the 12-sphere model of the pill particle C is constructed by filling spheres $O_7$ to $O_{12}$ with the mirror image of spheres $O_1$ to $O_6$ with respect to the xoy coordinate plane.

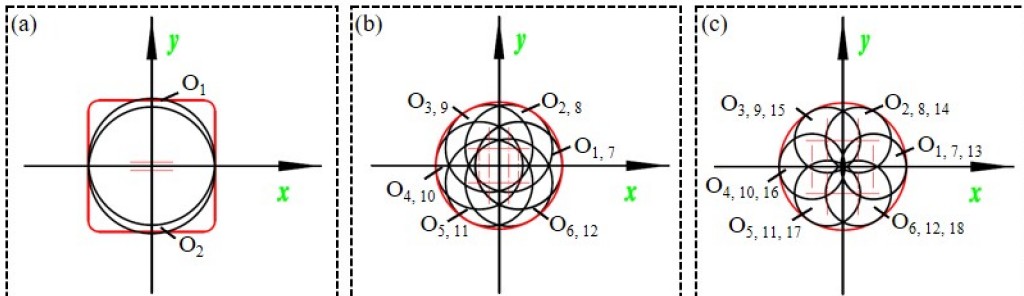

**Figure 4.** Diagram illustrating the procedure of filling a pill particle c: (**a**) A 2-sphere model; (**b**) A 12-sphere model; and (**c**) A 18-sphere model.

To further decrease inaccuracy, three layers of filled spheres were used to simulate the pill particles in the $z$-axis direction. At the point $(D/2-H/4, H/4)$ in the xoz coordinate plane, a sphere $O_1$ with radius $H/4$ and tangent to the cylindrical surface to the x axis fills the space; spheres $O_2$ to $O_6$ are formed by an array of spheres $O_1$ arranged at a $60°$ angle around the z axis in the oxyz coordinate system. The spheres $O_7$ to $O_{12}$ are filled with the mirror image of the spheres $O_1$ to $O_6$ in the xoy coordinate plane. The point $(D/2-H/4, 0)$ in the xoz coordinate plane is filled by a sphere $O_{13}$ with radius $H/4$ and tangent to the cylindrical surface to the x axis, and spheres $O_{14}$ to $O_{18}$ are generated by an array of spheres $O_{13}$ arranged at a $60°$ angle around the z axis in the oxyz coordinate system, thereby generating the 18-sphere model of the pill particle C shown in Figure 4c.

Figure 5 illustrates the 4-, 12-, and 20-sphere particle models of the pill particles A and B, the pill particle C's 2-, 12-, and 18-sphere particle models.

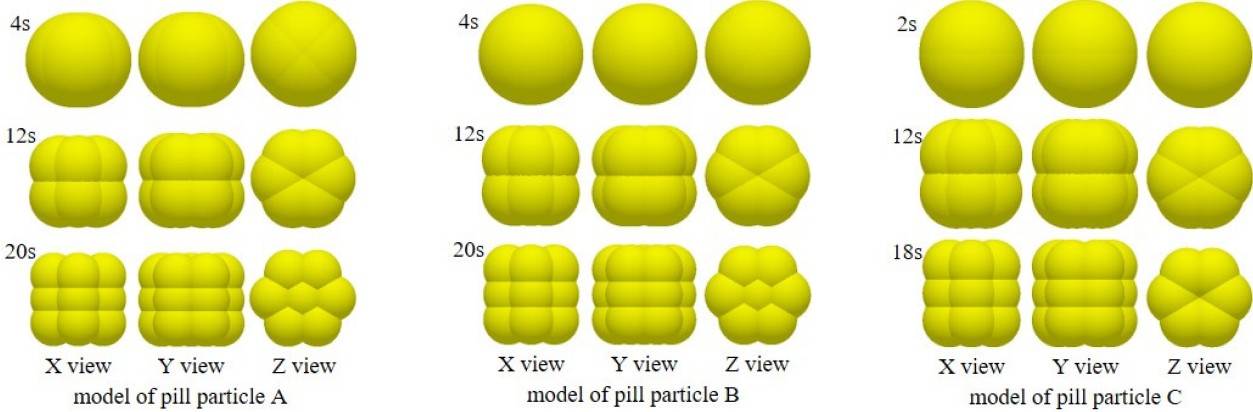

**Figure 5.** Three pill particle numerical models.

## 3. Experimental Verification and Simulation Analysis

The feasibility and effectiveness of the pill particle population modelling approach are shown in this study by a comparison of simulation findings for the bulk density and angle of repose tests to experiment data.

### 3.1. Simulation Model

In this paper, the Hertz–Mindlin contact model is used to simulate the bulk density test and the angle of repose test based on the material and physical properties of the pill particle. This contact model is one of DEM's most fundamental and well-known models, and it is commonly used to study the contact and mechanical behavior of non-viscous particles [27–30]. The motion of a particle is solved by the basic theories adopted by DEM, such as Newton's second law, Euler's equation, and dynamic relaxation method [30–33], given by

$$m_i \frac{d\mathbf{v}_i}{dt} = \sum_j (\mathbf{F}_{ij}^n + \mathbf{F}_{ij}^s) + m_i \mathbf{g} \tag{1}$$

and

$$I_i \frac{d\boldsymbol{\omega}_i}{dt} = \sum_j \left( \mathbf{R}_i \times \mathbf{F}_{ij}^s - \mu_r R_i \left| \mathbf{F}_{ij}^n \right| \hat{\boldsymbol{\omega}}_i \right) \tag{2}$$

where $\mathbf{v}_i$, $\boldsymbol{\omega}_i$ and $I_i$ are the translational and angular velocities, and moment of inertia of particle $i$, respectively. $\mathbf{R}_i$ is a vector running from the centre of the particle to the contact point with its magnitude equal to particle radius $R_i$. $\mu_r$ is the coefficient of rolling friction. $\mathbf{F}_{ij}^n$ and $\mathbf{F}_{ij}^s$ are the normal contact force and tangential contact force between particle $i$ and $j$, given by

$$\mathbf{F}_{ij}^n = \left( \frac{4}{3} E^* \sqrt{R^*} \delta_n^{\frac{3}{2}} - 2\sqrt{\frac{5}{6}} \beta \sqrt{S_n m^*} (\mathbf{v}_{ij} \cdot \hat{\mathbf{n}}_{ij}) \right) \hat{\mathbf{n}}_{ij} \tag{3}$$

and

$$\mathbf{F}_{ij}^s = \min \left( -S_t \delta_t - 2\sqrt{\frac{5}{6}} \beta \sqrt{S_t m^*} (\mathbf{v}_{ij} \cdot \hat{\mathbf{s}}_{ij}), \mu_s \mathbf{F}_{ij}^n \right) \hat{\mathbf{s}}_{ij} \tag{4}$$

where the equivalent Young's Modulus $E^*$ and the equivalent radius $R^*$ are defined as $E^* = \left[ \left(1 - v_i^2\right)/E_i + \left(1 - v_j^2\right)/E_j \right]^{-1}$ and $R^* = \left[ 1/R_i + 1/R_j \right]^{-1}$ with $E_i$, $v_i$, $R_i$ and $E_j$, $v_j$, $R_j$, being the Young's Modulus, Poisson ratio, and Radius of particles $i$ and $j$, respectively; $\delta_n$ is the normal overlap; the damping factor $\beta$, the normal stiffness $S_n$ and the equivalent mass $m^*$ are given by $\beta = -\ln e / \sqrt{\ln^2 e + \pi^2}$, $S_n = 2E^* \sqrt{R^* \delta_n}$ and $m^* = \left[ 1/m_i + 1/m_j \right]^{-1}$ with $e$, $m_i$, and $m_j$ being the coefficient of restitution and the mass of each particle in contact; $\mathbf{v}_{ij}$ is the relative velocity between particle $i$ and $j$; the unit vector $\hat{\mathbf{n}}_{ij}$ is calculated as $\hat{\mathbf{n}}_{ij} = (\mathbf{R}_i - \mathbf{R}_j) / |\mathbf{R}_i - \mathbf{R}_j|$; the tangential stiffness $S_t$ is given by $S_t = 8G^* \sqrt{R^* \delta_n}$ with $G^*$ being the equivalent shear modulus; $\delta_t$ is the tangential overlap; $\mu_s$ is the coefficient of static friction; $\hat{\mathbf{s}}_{ij}$ is the unit tangent vector.

### 3.2. Measurement and Calibration of Physical and Mechanical Parameters

The determination of simulation parameters has a significant impact on the stability and accuracy of the simulation calculation, according to the literature and preliminary research [34,35]. Hence, to verify the correctness of the simulation test findings, it is necessary to correctly test the physical and mechanical characteristics of the pill particles. The ASAE standard specified the Poisson's ratio of the pill particles [36]. The density of the pill particles was determined using the hydrometer technique [37]. The elastic modulus of the pill particles was determined using a compression test on an electronic universal testing equipment [38]. The coefficients of static friction between the pill particles and between the pill particles and the wall surface were determined using the slope technique [39] (ABS plastic, organic glass, and galvanized steel plate). The restitution coefficients between pill particles and between pill particles and the wall surface were determined using a high-speed camera and a drop test and a single pendulum test, respectively [40].

Due to the non-sphere surface solid nature of the pill particles, it is impossible to directly quantify the coefficients of rolling friction between the pill particles and between the pill particle and the wall surface. This research calibrates the simulation parameters using an angle of repose test and simulation in order to get them closer to their actual values. The findings indicate that the angle of repose is unaffected by the coefficient of rolling friction between the pill particle and the wall surface. As a result, the paper's first selection of the coefficient of rolling friction is between pill particle and ABS plastic, organic glass, and galvanized steel plate. The coefficient of rolling friction between the pill particles was then established by comparing the angle of repose test results to those obtained from the simulation. Table 1 contains the simulation parameters.

**Table 1.** Parameter selection for simulation.

| Parameters | Pill Particle | ABS Plastic | Organic Glass | Galvanized Steel |
|---|---|---|---|---|
| Density $\rho$, kg/m$^3$ | 1380 | 1050 | 1800 | 7865 |
| Poisson's ratio $v$ | 0.350 | 0.394 | 0.350 | 0.300 |
| Elastic modulus $E$, Pa | $1.100 \times 10^8$ | $3.189 \times 10^9$ | $1.300 \times 10^9$ | $7.900 \times 10^{10}$ |
| Restitution coefficient $e$ | 0.201 | 0.299 | 0.279 | 0.305 |
| Coefficient of static friction $\mu_s$ | 0.466 | 0.577 | 0.533 | 0.511 |
| Coefficient of rolling friction $\mu_r$ | 0.080 | 0.120 | 0.050 | 0.070 |

*3.3. Bulk Density Test and Simulation*

3.3.1. Bulk Density Test Setup

Bulk density is a critical statistic for evaluating the physical characteristics of granular materials, since it is directly connected to the shape and size of the particles. As a result, this paper employs the bulk density test to confirm the approach of pill particle modelling. The bulk density of the pill particles was determined in this research using a cylindrical measuring cylinder (organic glass) with a volume of 0.26 L. The procedure is as follows: first, the pill particles are released above the measuring cylinder and fall naturally; second, when the pill particles exceed the cylinder's height, the excess particles are scraped off with an organic glass scraper; and finally, the mass of the remaining particles in the cylinder is calculated, and the bulk density is calculated using the mass of the pill particles divided by the volume of the cylinder. As seen in Figure 6a, each set of tests was performed three times.

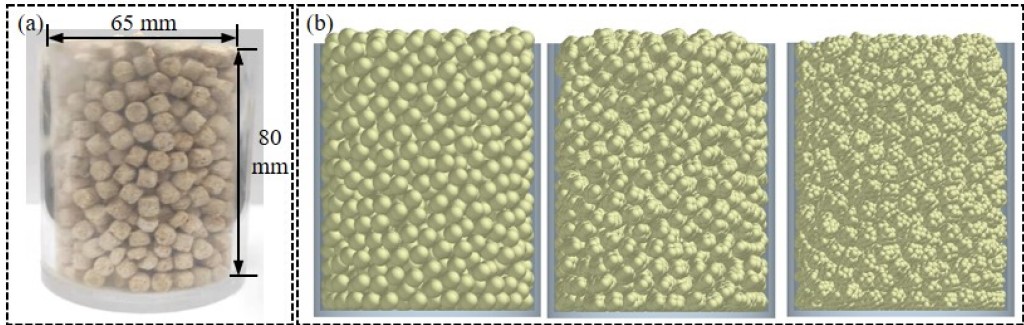

**Figure 6.** Bulk density test procedure and simulation analysis: (**a**) Bulk density test procedure; (**b**) Bulk density test simulation analysis of three pill particles.

3.3.2. Bulk Density Simulation Setup

The bulk density simulations of pill A (4-, 12-, and 20-sphere), pill B (4-, 12-, and 20-sphere), and pill C (2-, 12-, and 18-sphere) are performed using EDEM 2018 software with the Hertz–Mindlin (no-slip) contact model and the parameters listed in Table 1. The measuring cylinders used in the experiment are identical. When the simulation starts, a cylindrical region with a diameter of 20 mm and a height of 20 mm is placed as the particle factory 5 mm above the top of the measuring cylinder. Second, the pill particles are formed in the particle factory and gradually accumulate in the cylinder until they reach the cylinder's maximum height. When the pill particles have stabilized, the scraper is used to remove any remaining particles. Finally, the mass of the particles left in the measuring cylinder is determined. Each series of tests was done three times; the bulk density test simulation analysis is given in Figure 6b.

*3.4. Angle of Repose Test and Simulation*

3.4.1. Angle of Repose Test Setup

The angle of repose is also a critical parameter for assessing the physical and mechanical characteristics of granular materials, as it is directly connected to the form, size, and friction coefficient of the particles. As a result, the angle of repose test is utilized to confirm

the approach of pill particle modelling. The lifting technique is utilized in this article to examine the pill particle stacking process. The following are the test steps: To begin, a lifting cylinder (ABS plastic) with a diameter of 65.7 mm and a height of 150 mm is inserted into the center of a flat plate (galvanized steel) with a side length of 500 mm, and 0.5 kg of pill particles are poured into the cylinder, maintaining a flat surface on the particles; second, the cylinder is lifted upwards at a speed of 200 mm/min, causing the pill particles to flow out of the cylinder and make contact with the galvanized steel plate; finally, the particles are stabilized, forming a conical pile, and the angle of repose is determined using an image processing method. As seen in Figure 7a, each set of tests was performed three times.

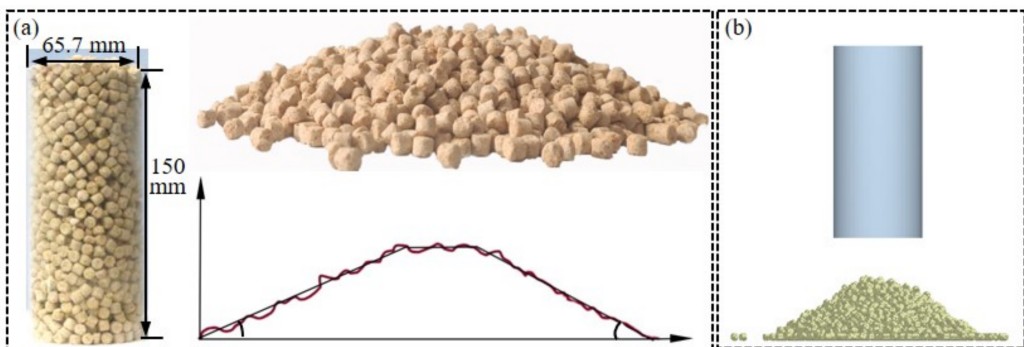

**Figure 7.** Angle of repose test procedure and simulation analysis: (**a**) Angle of repose test procedure and image processing method; (**b**) Angle of repose test simulation analysis.

3.4.2. Angle of Repose Simulation Setup

Angle of repose simulations of various pill particle models are carried out using EDEM 2018 software with the Hertz–Mindlin (no-slip) contact model and the parameters listed in Table 1. The test devices are identical to those used throughout the experiment. When the simulation starts, a cylindrical region with a diameter of 60 mm and a height of 20 mm is set up as the particle factory, 5 mm above the cylinder's top. Second, 0.5 kg of nearly 3000 pill particles are formed in the particle factory and gradually accumulate in the cylinder until they reach the cylinder's maximum height. Once the pill particles are stable, the cylinder is elevated at a 200 mm/min pace. Finally, after the pill particles have ceased to flow, a conical particle heap is produced, and the angle of repose on both sides is determined using an image processing technique. Each pair of tests is done three times; the angle of repose test simulation analysis is displayed in Figure 7b.

## 4. Results Analysis and Discussion

### *4.1. Results Analysis and Discussion of Bulk Density Test*

4.1.1. Bulk Density Test Results Analysis

Figure 8 illustrates the fluctuation in the bulk density test simulation results with the number of filled spheres for various pill particle models, and the green region in the figure depicts the bulk density test results and standard deviation.

The change in the bulk density of the pill particles A as a function of the number of filled spheres is seen in Figure 8a. As the number of filled spheres grows from 4 to 20, the particle bulk density simulation results first increase and then decline, which is more consistent with the experiment data. Further research reveals that the relative errors of the simulation and experiment data are 1.11%, 0.06 percent, and 2.93%, respectively, for the number of filled spheres of 4, 12, and 20. As can be seen, the pill particle model's bulk density is affected by the number of filled spheres.

The change in the bulk density of the pill particles B as a function of the number of filled spheres is seen in Figure 8b. As seen in the image, the results of the discrete bulk density simulation grow and then decline as the number of filled spheres increases. The relative errors between simulation and experiment data were 2.10%, 1.28%, and 3.89%, respectively, for the number of filled spheres of 4, 12, and 20.

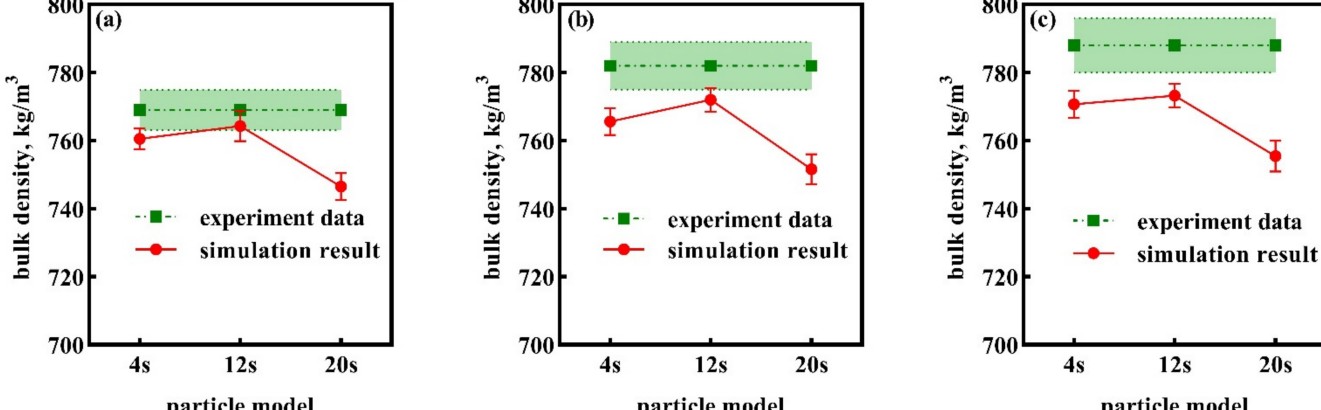

**Figure 8.** The findings of a simulation of the bulk density test of a pill particle model with varying numbers of filled spheres: (**a**) Results of the simulation for pill particle A; (**b**) Results of the simulation for pill particle B; and (**c**) Results of the simulation for pill particle C.

The fluctuation in the bulk density of the pill particles C as a function of the number of filled spheres is seen in Figure 8c. When shown in the figure, as the number of filled spheres increases, the particle bulk density simulation results initially rise and then decline, eventually approaching the experimental data, which is compatible with the changing trend of pill particles A and B. The relative errors between the simulation and experiment data were 2.20%, 1.88%, and 4.14%, respectively, for the number of filled spheres of 2, 12, and 18.

### 4.1.2. Bulk Density Test Discussion

According to the bulk density simulation and test results, when the number of filled spheres is 4 or 2, the space around the model is more defective than the actual pill particles, but the simulation results are closer to the test data due to the particles' reduced self-locking and increased sliding and rolling. The particle model becomes more accurate as the number of filled spheres increases, and the simulation results progressively converge to the test findings. When the number of filled spheres is 20 or 18, the filled spheres in the center of the particles increase their frictional impact, which has an influence on their movement and reduces the simulation results. The simulation findings validate the model suggested in this paper's accuracy for varied pill particle populations.

Furthermore, because the bulk density test is relatively simple, the simulation process takes less time, with different models taking less than 1 h to simulate. In conclusion, based on the simulation findings of the pill particle bulk density test, other modelling techniques, with the exception of the pill particle A and B 20-sphere models and the pill particle C 18-sphere model, have a poor correlation with the experiment results.

### 4.2. Results Analysis and Discussion of Angle of Repose

#### 4.2.1. Angle of Repose Test Results Analysis

Figure 9 illustrates the relationship between the angle of repose test simulation results and the number of filled spheres for various pill particle models, while the green region in the figure depicts the angle of repose experiment findings and standard deviation.

Figure 9a illustrates the relationship between the angle of repose of pill particle A and the number of filled spheres in the simulation. As seen in the image, the particle angle of repose simulation results steadily rise as the number of filled spheres grows from four to twenty. The simulation findings for the four-sphere model are insignificant and do not match the experiment data. The angle of repose simulation findings for the 12- and 20-sphere models are within the standard deviation of the experiment data and near to the mean, however the change in the number of filled spheres in the pill particles model resulted in considerable angle of repose variations.

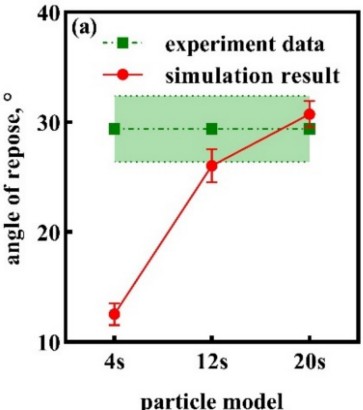 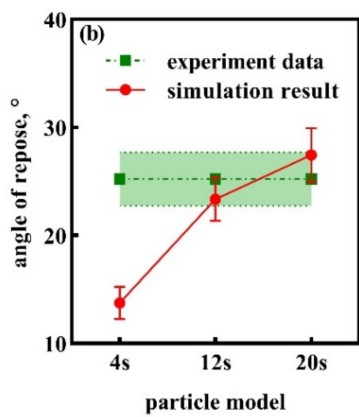 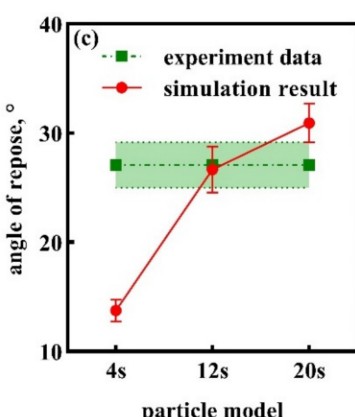

**Figure 9.** Angle of repose test results for the pill particle model as a function of the number of filled spheres: (**a**) Results of the simulation for pill particle A; (**b**) Results of the simulation for pill particle B; and (**c**) Results of the simulation for pill particle C.

Figure 9b illustrates the relationship between the angle of repose of pill particle B and the number of filled spheres in the simulation. As seen in the image, the particle angle of repose simulation results steadily rise as the number of filled spheres increases. The simulation findings for 12 and 20 filled spheres are within the standard deviation of the experiment data.

The angle of repose of the pill particle C is shown in Figure 9c as a function of the number of filled spheres. As shown in the image, the particle angle of repose simulation results steadily increases as the number of filled spheres increases, which is similar with the pattern seen for pill particles A and B. When the number of filled spheres is 12, the smallest relative error between the simulation and experiment data is 1.59%.

### 4.2.2. Angle of Repose Test Discussion

The test and simulation findings demonstrate that the intricacy of the pill's particle form results in the particles self-locking throughout the piling process. The number of particle-filled spheres has a significant influence on the geometry of the particle model, which in turn has an effect on the angle of repose simulation results. When the number of particle-filled spheres is small, the sphericity of the particles rises and the roughness and self-locking of the particles decrease, increasing the particles' sliding and rolling. As the number of filled spheres rises, the particle form eventually converges to that of genuine pill particles, increasing both the particles' self-locking behavior and the pile's angle of repose. The simulation findings further demonstrate the correctness of the approach for modelling pill particle populations suggested in this work.

Moreover, since the angle of repose test is slightly more complex than the bulk density test, the simulation time increases as the number of filling spheres increases, but the total simulation time for all models is less than 5 h. To summarize, based on the computational efficiency and the analysis of the simulation results for the angle of repose test of the pill particles, the simulation results for the 12- and 20-sphere models of pill particle A and B, as well as the 12-sphere model of pill particle C, are within the standard deviation of the experiment results. According to a comprehensive analysis of the bulk density and angle of repose test simulation results, the 12-sphere models of pill particles A, B, and C are close to the actual pill particle shapes, and the simulation results have small errors in comparison to the experimental data, making them suitable for simulating the mechanics behavior of particles and their flow processes during the actual pill discharging process.

### 5. Example Applications

On the basis of the preceding work, the applicability of the suggested modelling technique for pill particle population and its practical use are shown by simulation results of the pill particle discharge process.



### 5.1. Pill Discharging Process Device

The research is based on a self-designed pill discharge mechanism, which includes the primary functioning components depicted in Figure 10a, including the material box, the pill wheel, and the frame. The material box is stamped from a galvanized steel plate, and its conical top half-angle shape is inspired by the friction properties of pill particles. The pill wheel is made of ABS technical plastic alloy and has eight identical medication feeding grids, each measuring 100 mm in length.

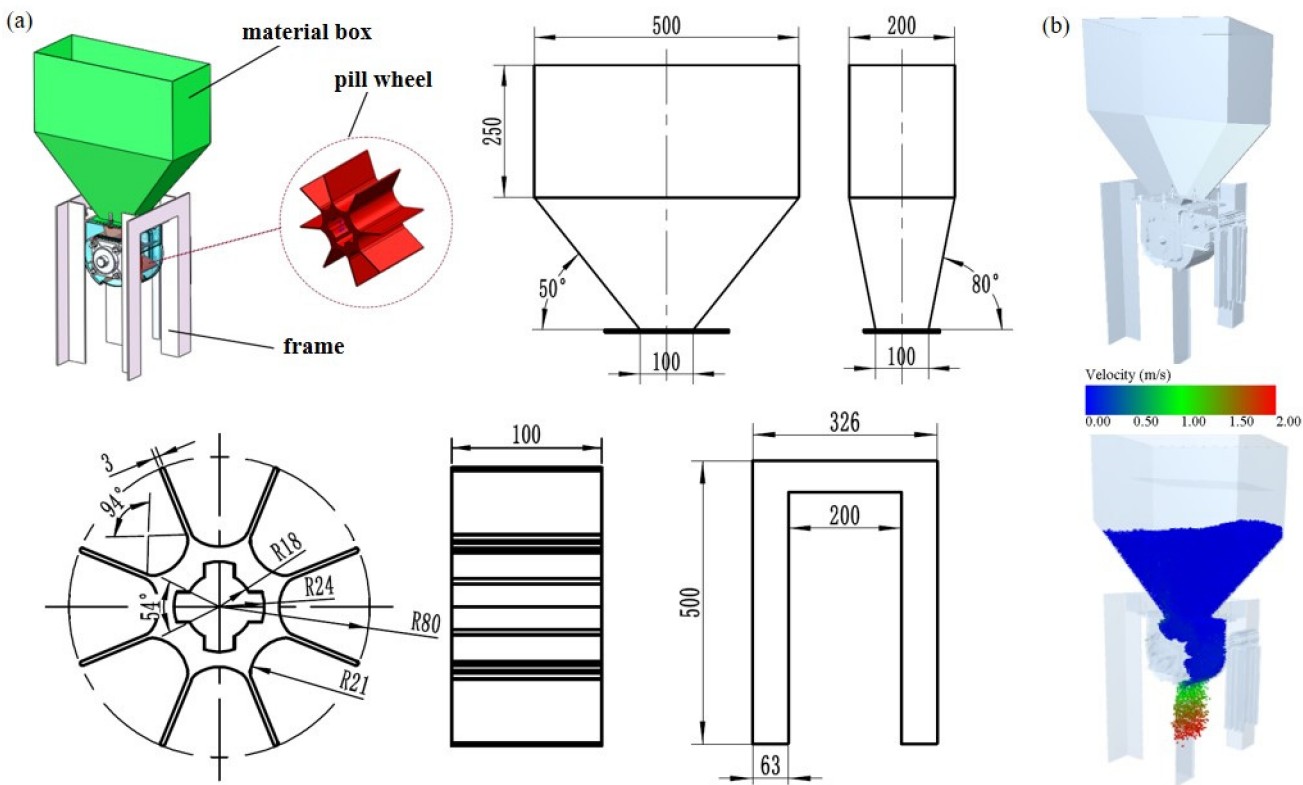

**Figure 10.** Test device and simulation analysis of the pill discharge process: (**a**) Test device for the pill discharge process; (**b**) DEM model of the pill discharge device and simulation analysis of the pill discharge process.

### 5.2. Simulation Setup of Pill Discharging Process

The pill particle discharge processes are simulated using EDEM 2018 software with the Hertz–Mindlin (no-slip) contact model and the parameters listed in Table 1. The particle models of 12-sphere pills A, B, and C are utilized in proportions of 35%, 35%, and 30%, respectively. Figure 10b depicts the DEM model of a pill particle discharge device. When the simulation starts, a box area with a length of 400 mm and a width of 150 mm is placed as the particle factory 5 mm above the top of the work bin. Second, 10 kg of nearly 60,000 pill particles are produced in the particle factory and gradually accumulate in the hopper. When the pill particles reached a state of stability, the pill wheel started to spin (i.e., 10, 15, and 30 rpm). Finally, the mass of particles ejected by the pill wheel spinning one grid every 10 s is determined. Thus, the discharge capacity and variation coefficient of pill particles are determined. Each pair of tests is done three times; Figure 10b depicts a simulation study of the pill particle discharge process.

### 5.3. Results Analysis and Discussion of Discharge Process

The mass of the pills discharged and its variation coefficient at various rotating speeds throughout the pill release process are shown in Figure 11.

Figure 11a illustrates the simulation findings for the time-dependent change in pill particle discharging at various rotation speeds. As seen in the figure, the mass of discharg-

ing pill particles grows almost linearly with the simulation duration at various rotating speeds. The pill wheel's speed was raised from 10 to 30 rpm, and the amount of pills discharged steadily rose, while the filling mass of the medication feeding grid was reduced from 0.25 to 0.23 kg.

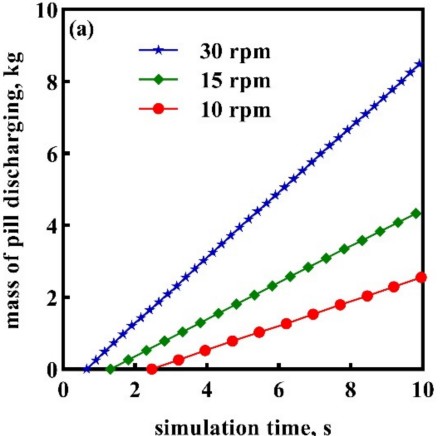 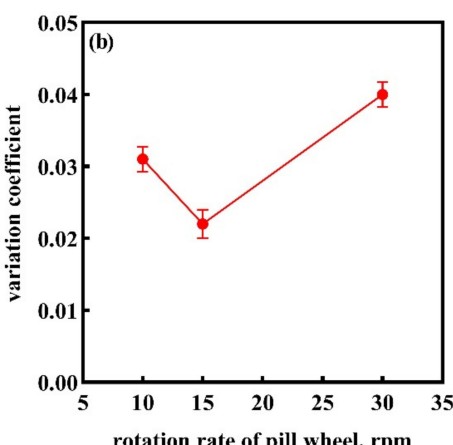

**Figure 11.** The simulation results for pill particle discharging mass and its variation coefficient at various rotation speeds during the pill discharging process: (**a**) Simulation results for pill particle discharging mass; (**b**) Simulation results for pill particle discharging mass variation coefficient.

Figure 11b illustrates the simulation findings for the change in the discharge efficiency of pill particles as a function of the pill wheel's rotation speed. When seen in the figure, the mass variation coefficient drops from 0.03 to 0.02 as the rotation speed of the pill wheel increases from 10 to 15 revolutions per minute. When the rotation speed is increased to 30 rpm, the variation coefficient of the pill discharge mass rises from 0.02 to 0.04. As can be observed, the variation coefficient of the pill wheel's rotation speed has a substantial influence on the mass of pill particles discharged and their stability.

As a result of the simulation findings, it is clear that the rotation speed of the pill wheel is the most critical operating parameter of the pill discharging device, as it has a significant effect on the mass and stability of the pills discharged. When the rotation speed is low, it has a negligible influence on the filling of the pill discharging grid, while increasing the rotation speed marginally enhances the discharging capability. When the rotation speed exceeds a particular threshold, it has an influence on the filling mass of the pill discharging grid, which in turn has an effect on the discharging stability, with the effect steadily rising as the rotation speed increases. The simulation findings in this work are compatible with those in the literature on fertilizer discharge processes [5,41], demonstrating the applicability of the pill particle population modelling approach presented in this paper and its practical use. For further in-depth investigations of the pill discharging process, it offers an accurate simulation model and theoretical foundation for optimizing the structural parameters, dimensions parameters, and operating parameters of the pill discharging device.

## 6. Conclusions

This paper uses self-developed anticorrosive pill particles as the research object, proposes a pill particle population modelling method based on DEM by testing and analyzing the pill particles' shape and size parameters, and verifies the modelling method's feasibility, efficacy, and applicability via experimental research and simulation analysis. The following are the study's major conclusions:

(1)  The particle shape and size parameters were evaluated and analyzed to approximate the cylindrical shape of the pill particles, and the particle population was classified into pill particles A (5.4 mm), B (5.8 mm), and C (6.2 mm) based on their height, with the mass ratio of particles accounting for 35%, 35%, and 30%, respectively.

(2)  This work proposes a population modelling approach for pill particles based on DEM. Multi-sphere particle models were created for pill particle A (4, 12, and 20 spheres), pill particle B (4, 12, and 20 spheres), and pill particle C (2, 12, and 18 spheres). It serves as a guide for modeling cylindrical and irregular particles.

(3)  Using the bulk density and angle of repose tests as examples, the pill particle population modelling approach was utilized to deduce the mechanism by which the number of pill particle-filled spheres affects the particle accumulation process and flow behavior. By comparing the simulation findings to the test data, the feasibility and efficacy of the pill particle population modelling approach were established.

(4)  Using the independently built pill discharging device as an example, the 12-sphere model of pill particles A, B, and C was utilized to deduce the process by which the wheel's rotation speed affects the pill discharging performance. The method's applicability and practical use were shown by assessing the simulation results of the pill discharging process and establishing the groundwork for future improvements of the pill discharging device.

**Author Contributions:** Conceptualization, D.L. and Y.Y.; methodology, D.L. and Y.Y.; validation, C.Q. and Y.L.; resources, C.Q.; writing—original draft preparation J.S.; writing—review and editing, D.L.; supervision, Z.Y. and X.W.; project administration, J.W. All authors have read and agreed to the published version of the manuscript.

**Funding:** The research was financially supported by Scientific Research Project of the CNOOC EnerTech-Drilling & Production Co., grant number GCJSXMHT-T2101.

**Institutional Review Board Statement:** Not applicable.

**Informed Consent Statement:** Not applicable.

**Data Availability Statement:** Not applicable.

**Conflicts of Interest:** The authors declare no conflict of interest.

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
