# Peer review of "Modelling Method and Application of Anti-Corrosion Pill Particles in Oil and Gas Field Wellbore Casing Annulus Based on the Discrete Element Method"

_processes, doi:10.3390/pr10061164_

Round 1

Reviewer 1 Report

This study uses a self-developed anti-corrosion pill particle as the research object and develops the pill particle population modelling method in order to optimize the anti-corrosion process of oil and gas wellbore casing annulus.

Thematically the work is interesting for the researchers and professionals and the proposed manuscript is relevant to the scope of the journal.

I found it appropriate for publication in the Processes journal, but only after some modifications and clarification from the Authors.

The title is a clear representation of the manuscript's content. 

The overall organization and structure of the manuscript are appropriate. The paper is well written and the topic is appropriate for the journal.
The aim of the paper is well described and the discussion was well approached, its results and discussion are correlated to the cited literature data.

The literature review is comprehensive and properly done.

The novelty of the work must be more clearly demonstrated.

The significance of the Work: Given the large number of analyzed data, this is an interesting study with a possible significant impact in this area.

Statistical interpretation of the analytical data must be more properly presented. The verification of the model should be performed. 

Other Specific Comments: The work is properly presented in terms of the language. The work presented here is very interesting and well done, it is presented in a compact manner.
In general, there are no doubtful or controversial arguments in the manuscript. The methodology applied in the research is presented in clear manner, so that it is repeatable by other authors.

Some specific comments are given bellow:

  1. please explain more about the particle trajectories? Equations 1 and 2 explain these data? Do particle properties and equipment geometry affect the results?
  2. perhaps somewhat better explanation regarding the simulation setup of pill discharging process could be provided? What software was used in simulation? EDEM?
  3. how many particles were used in the model?
  4. what about the particle to wall contact? Was it adjusted to match experimental and numerical results? Is there any difference considered between the parameters used for the particle/particle contact and particle/wall contact? Perhaps a better explanation of coefficients presented in Table 2 should be presented?
  5. Fig. 1 and Table 1 should be combined, presently the same information is shown in both cases.
  6. There are no references in Simulation model section? These equations are well-known and frequently mentioned in the literature?
  7. Figures presenting material and/or material simulation and/or experimental setup are presented with no dimensions shown, with no model boundaries explained? 

Reviewer 2 Report

This paper proposes a pill particle population modelling method based on DEM for the anti-corrosion process of oil and gas wellbore casing annulus. The paper is well written and presented, and the results obtained are intersting. 

Some minor comments with regard to the paper:

i) In the introduction section, no mention of references [6-10].

ii) Include more updated references on the multi-sphere method, for this area of research.

iii) Section 3.1 - lines 192-202 re-write the maths parameters/ equations so as to be aligned with the text.

iv) For the pill particles, the number used are 2s, 4s, 12s, and 18s and 20s. Any reason this? Can this approach be applied to irregular shapes?

v) In some previous work, it was indicated that using more than six multi-spheres would slow the simulation significantly, thus increasing computation time.  In the present work, what is the simulation time? is it significant?

Round 2

Reviewer 1 Report

The authors managed to improve the quality of the Manuscript according to the Reviewer's comments.

I suggest the Editor to accept the Manuscript in the presented form for possible publication in the Processing journal.

Author Response

Dear Reviewers:

Thank you for the reviewers’ comments concerning our manuscript entitled “Modelling method and application of anti-corrosion pill particles in oil and gas field wellbore casing annulus based on DEM” (processes-1731557). Those comments are all valuable and very helpful for revising and improving our paper, as well as the important guiding significance to our researches.